# The mediating role of resilience on the association between family satisfaction and lower levels of depression and anxiety among Chinese adolescents

Beizhu Ye[1], Joseph T. F. Lau[2,3,4], Ho Hin Lee[2], Jason C. H. Yeung[2], Phoenix K. H. Mo[2]*

**1** Department of Social Medicine and Health Management, School of Public Health, Zhengzhou University, Zhengzhou, China, **2** Center for Health Behaviours Research, JC School of Public Health and Primary Care, Faculty of Medicine, The Chinese University of Hong Kong, Hong Kong, Hong Kong, **3** School of Mental Health, Wenzhou Medical University, Wenzhou, China, **4** Zhejiang Provincial Clinical Research Center for Mental Disorders, The Affiliated Wenzhou Kangning Hospital, Wenzhou, China

* phoenix.mo@cuhk.edu.hk

**Data Availability Statement:** Data cannot be shared publicly because of participants' privacy. Data are available from the Chinese University of

## Abstract

### Purpose

This study aimed to explore the association between family satisfaction, resilience, and anxiety and depression among adolescents, and the mediating role of resilience in these relationships.

### Methods

A cross-sectional study was conducted among grade 8 to 9 students from 4 secondary schools in Hong Kong. A total of 1,146 participants completed the survey.

### Results

Respectively 45.8% and 58.0% of students scored above the cut-off for mild anxiety and mild depression. Results from linear regression analyses showed that family satisfaction was positively associated with resilience, and both family satisfaction and resilience were and negatively associated with anxiety and depression. The mediating effects of resilience on the relationship between family satisfaction and anxiety/ depression (26.3% and 31.1% effects accounted for, respectively) were significant.

### Conclusions

Both family satisfaction and resilience have important influence on adolescent mental health. Interventions that seek to promote positive family relationships and resilience of adolescents may be effective in preventing and reducing anxiety and depression symptoms among adolescents.

Hong Kong Survey and Behavioral Ethics Committee (contact via sbrec@cuhk.edu.hk) for researchers who meet the criteria for access to confidential data.

**Funding:** The author(s) received no specific funding for this work.

**Competing interests:** The authors have declared that no competing interests exist.

## Introduction

Mental health problems, with most of them having a relatively young onset, have significantly affected children and adolescents worldwide. Anxiety and depression are the most common mental health problems reported, and they are the top ten causes of disability-adjusted life-years for younger population aged 10–24 years [1–3]. A sharp increase in anxiety and depression has been observed over the past few decades [4, 5]. In Hong Kong, a representative study among 9,518 secondary school students showed that the level of depression was at moderate or severe level as measured by the Center for Epidemiological Studies-Depression (CES-D) [6]. Another study among 3,136 secondary school students in Hong Kong showed that respectively 54.3% and 65.8% of males and females scored above the cut-off for mild depression, as measured by the CES-D [7].

Anxiety and depression are associated with significant adverse consequences among adolescents, such as substance abuse, poor physical health, underachievement in schools, harmful social outcomes, subsequent depression in later life, and higher risk of suicide [8, 9]. Moreover, they may induce long-term effects throughout life, and even affect the mental health of offspring [10–13]. Accounting for the largest population in the world, China suffers from a huge burden of mental health problems. Understanding the factors underlying the occurrence of mental problems is greatly warranted.

Adolescents are at a critical period of physical and mental development. They constantly strive for independence but simultaneously may encounter many physical and psychosocial challenges as they cope with the tremendous stressors in school and social lives. Studies have shown that adolescents' anxiety and depression are subject to the influence of various individual and social factors [14–18]. Among them, the influence of family on adolescent mental health is instrumental [19–21]. Specifically, family satisfaction, which measures the extent to which one is satisfied with one's family's overall functioning and relationships, is one of the most influential aspects of life satisfaction. Studies have shown that higher family satisfaction leads to lower levels of worry, anxiety, and depression in adolescents [22]. Both cross-sectional and longitudinal studies have proven that family satisfaction was a significant protective factor of anxiety and depression among adolescents [23–25]. One study in South America found that family satisfaction and communication explained 18% of the variance in child anxiety [26].

Despite the strong link between family satisfaction and mental health problems established in the literature, very little is known about how the association between family satisfaction and mental health problems can be conceptualized. The present study proposed that resilience may be an important mediator that help to clarify the association between family satisfaction and anxiety/ depression. Psychological resilience commonly pertains to the capacity of an individual to survive and adapt to difficult situations [27, 28]. Individuals who are resilient possess the ability to effectively cope with adversities and stress while maintaining normal mental and physical functioning [29, 30]. A dynamic systems perspective of resilience has been reflected recently, conceptualizing it as the capacity of dynamic systems to adapt to adversities and threats [31]. Resilience was found to be strongly associated with mental health in children and adolescents, contributing to positive outcomes, and serving as protective factor within individuals' well-being [32, 33]. The pivotal role of resilience in saving individuals from mental disorders amid violent and life-threatening events was also demonstrated in the literature [34, 35]. Furthermore, school-based resilience-focused interventions were effective in reducing depressive symptoms, internalizing problems, externalizing problems, and general psychological distress among children and adolescents [36, 37].

Family is also a fundamental source of support and an influential factor of resilience for adolescents. Adolescents who have a positive family environment might be more likely to

acquire the skills in coping with stressors and establishing a positive personal development. Studies have reported that positive family factors, such as lower levels of parental psychological control and maternal overprotection, high levels of maternal warmth, greater family cohesion, and easier temperament were associated with greater odds of resilience in youths [38, 39]. Masten's multisystem resilience approach also included a multitude of crucial family attributes, such as sensitive caregiving, family cohesion, sense of belonging, and family management in facilitating adolescent resilience [31]. Nevertheless, very few studies have examined the role of family satisfaction on resilience in the Chinese context.

The consistent empirical evidence established between resilience, family satisfaction and mental health suggested that resilience could be a potential mediator of the association between life satisfaction and mental health. Meanwhile, extant theories also lent support to the potential mediating role of resilience. According to the Resiliency Theory, two types of promotive factors operate to ameliorate risk factors and mental ill health: positive factors within individuals and resources available outside the individuals [40, 41]. In this regard, family satisfaction could serve as an important positive resource that promotes resilience, which further buffer the negative effects of stress of mental illness. Growing evidence on the mediating role of resilience on mental health has been reported in literature [42–44]. For example, a study among 2,925 medical students in China found that resilience mediated the relationship between personality traits and anxiety [45]. Another recent study among 1,845 adolescents in China also found that resilience mediated the link of subjective family socioeconomic status with life satisfaction [46]. However, to the best of our knowledge, the possible mediating role of resilience in the association between family satisfaction on anxiety and depression has not yet been investigated.

Considering the significant mental health problems documented among Chinese adolescents, there is an urgent need to understand the factors associated with their mental health so that appropriate interventions could be designed. With respect to the indispensable role of family factors and resilience on adolescent mental health, it is expected that findings of the present study would provide important directions and implications on the design of mental health interventions for Chinese adolescents. The present study aimed to explore the association between family satisfaction and resilience on mental health problems (i.e. anxiety and depression) among adolescents in Hong Kong. It also examined whether resilience mediated the relationship between family satisfaction and anxiety/ depression. We hypothesized that family satisfaction would be associated with higher levels of resilience, which in turn, would be associated with lower levels of anxiety/ depression among adolescents.

## Methods

### Study design

A cross-sectional study was conducted among secondary school students in Hong Kong. Invitation letters were sent to all secondary schools of Hong Kong, four replied and showed interest in taking part in the study. Discussions were made with the school management team, it was agreed that secondary two and three students (i.e. grade eight and nine) would be invited as secondary one students have just transited to secondary school and secondary four to six students would need to prepare for public examinations. An opt-out parental informed consent was used, in which parents were asked to return a signed form if they refused to let their students take part in the survey. No parents returned such forms. In a classroom setting and with the absence of teachers, research assistants briefed the students the objectives of the study and the logistics that would be involved. Participants were informed that their participation was voluntary and return of the questionnaire implied informed consent. They were also

guaranteed that the data would be kept confidential and only be accessible by the researchers of the team. Students then self-administered the questionnaire which took about 20 minutes to complete. The questionnaire was sent to a total of 1,185 students, and all of them returned the questionnaire. Ethics approval was obtained from the Survey and Behavioral Ethics Committee of the institution of the Chinese University of Hong Kong.

As a measure of quality control, 39 cases with at least one scale having more than 20% of its items missing were excluded from data analysis. The final sample consisted of 1,146 participants.

## Measures

*Anxiety* was assessed by the 7-item General Anxiety Disorder (GAD) Scale [47]. It has been validated among Chinese adolescents [48]. Each item was rated on a 4-point scale (0 = not at all to 3 = almost all of the time). The total score ranges from 0 to 21, and higher scores indicate more severe anxiety symptoms. The Cronbach's alpha was 0.941 in the present study. The GAD scores of $\geq 5$, 10, and 15 are defined as mild, moderate, and severe anxiety respectively.

*Depression* was measured by the Chinese version of 20-item CES-D [49]. It has been validated among Chinese adolescents [50, 51]. Participants were inquired about how often they experienced the feelings in the past week. Each item is rated on a 4-point scale (0 = rarely or none of the time to 3 = almost or all of the time). The total score ranges from 0 to 60, and higher scores indicate more severe depressive symptoms. The Cronbach's alpha was 0.905 in the present study. The CES-D scores of $\geq 16$, 21 and 25 are defined as mild, moderate, and severe depression respectively.

*Family satisfaction* was measured by the 10-item Family Satisfaction Scale, which measures one's level of satisfaction with family overall functioning and relationships [52]. It has been used among adolescents [53, 54]. Each item is rated on a 5-point scale (1 = very dissatisfied to 5 = very satisfied). The total score ranges from 10 to 50, and higher scores indicate greater satisfaction with the family. The Cronbach's alpha was 0.962 in the present study.

*Resilience* was assessed by the 25-item Connor-Davidson Resilience Scale [55]. It has been validated among Chinese adolescents [56]. Participants were asked how often they experienced various situations on a 5-point scale (1 = not at all to 5 = almost all of the time). The total score ranges from 25 to 125, and higher scores indicate higher level of resilience. The Cronbach's alpha was 0.939 in the present study.

*Socio-demographic variables* included age (as a continuous variable), sex (male / female), grade (eighth / ninth grade), and housing type (public housing / Home Ownership Scheme (HOS) / private housing / others).

## Statistical analysis

The missing values from all of the items ranged from 0.1% to 0.4%, which were imputed by using the modal value of the other items of the same scale. Descriptive statistics were presented. Continuous variables are presented as means ± standard deviations, and categorical variables are shown as percentages. Correlation between the variables under studied was calculated. To test the association between resilience and family satisfaction on anxiety/ depression, a series of hierarchical linear regression models were performed. Model 1 included socio-demographic variables only; model 2 added family satisfaction; and model 3 added resilience. Standardized regression coefficient ($\beta$), F, $R^2$ and $R^2$-changes ($\Delta R^2$) for each regression model were provided.

In addition, asymptotic and resampling strategies developed by Preacher and Hayes [57] were employed to test the significance of the mediating role of resilience on the associations

between family satisfaction and anxiety/ depression. In the regression equations, family satisfaction was modeled as predictors, anxiety/ depression as the dependent variable, resilience as the mediator, and socio-demographic variables as covariates. The path coefficients a (family satisfaction to resilience), b (direct effects of resilience on anxiety/ depression), a*b products (indirect effects of family satisfaction on anxiety/ depression through resilience), c (total effects of family satisfaction on anxiety/ depression) and c' (direct effects of family satisfaction on anxiety/ depression) were presented. We applied bootstrapping methods to obtain a 95% confidence interval (CI) for the mediated effects. The bootstrap estimate was based on 5,000 bootstrap samples, and the bias-corrected and accelerated 95% confidence interval (BCa95%CI) of indirect effect (a*b product) excluding 0 indicated a significant mediating role. We conducted all analyses using SPSS 22.0 (SPSS Inc., Chicago, IL). Statistical tests were two-tailed, and differences were considered significant when $p < 0.05$.

## Results

### Socio-demographic characteristics and mental health status

The socio-demographic characteristics and mental health status of the participants are shown in Table 1. Average age was 13.58 years old (SD = 0.88), ranging from 11 to 18. Among the 1,146 students, 644 (56.89%) were males.

The average score of depression and anxiety of the current sample revealed a concerning mental health issue. Respectively 45.8% (49.2% for female and 42.9% for male, $p = 0.022$) and 58.0% of students (61.9% for female and 55.1% for male, $p = 0.034$) scored above the cut-off for mild anxiety and mild depression.

### Correlations among major variables

Results of Pearson correlation analysis are shown in Table 2. Family satisfaction was positively correlated with resilience (r = 0.42, $p < 0.001$) and negatively correlated with anxiety (r = -0.29, $p < 0.001$) and depression (r = -0.38, $p < 0.001$). Resilience was negatively correlated with anxiety (r = -0.28, $p < 0.001$) and depression (r = -0.41, $p < 0.001$). The results were consistent to our hypothesized relationship among the variables.

### Linear regression model for anxiety

Results of the hierarchical linear regression models of anxiety are presented in Table 3. Among the socio-demographic characteristics, female gender was positively related to anxiety ($\beta$ = 0.07, $p < 0.05$). Also, living in private houses was negatively associated with depression ($\beta$ = -0.08, $p < 0.05$). After controlling for these socio-demographic characteristics, family satisfaction ($\beta$ = -0.28, $p < 0.001$) was negatively associated with anxiety in Model 2. It remained significant ($\beta$ = -0.20, $p < 0.001$) after resilience ($\beta$ = -0.18, $p < 0.001$) was added in Model 3. Overall the model explains 12% of the variance in anxiety.

### Linear regression model for depression

Results of the hierarchical linear regression models of depression are presented in Table 3. Similarly, among the socio-demographic characteristics, female gender was positively related to anxiety ($\beta$ = 0.12, $p < 0.001$). Also, compared with students living in public housing, those living in The Home Ownership Scheme (HOS) ($\beta$ = -0.09, $p < 0.01$), private houses ($\beta$ = -0.11, $p < 0.01$), and others ($\beta$ = -0.06, $p < 0.05$), were negatively associated with depression. After controlling for socio-demographic characteristics, family satisfaction was negatively associated with depression ($\beta$ = -0.37, $p < 0.001$) in Model 2. It remained significant ($\beta$ = -0.25, $p < 0.001$)

**Table 1. Background characteristics of participants (N = 1,146).**

| Variable | N (%) | M (SD) |
|---|---|---|
| Age in years (Mean, SD) | 13.58 (0.88) | |
| Sex | | |
| Male | 644 (56.89) | |
| Female | 488 (43.11) | |
| Grade | | |
| Eighth grade | 546 (47.64) | |
| Ninth grade | 600 (52.36) | |
| Housing type | | |
| Public housing | 434 (38.27) | |
| Home Ownership Scheme (HOS) | 198 (17.46) | |
| Private housing | 478 (42.15) | |
| Others | 24 (2.12) | |
| Anxiety score | | 5.04 (5.30) |
| Normal | 621(54.19) | |
| Mild | 324(28.27) | |
| Moderate | 126(10.99) | |
| Severe | 75(6.54) | |
| Depression score | | 19.37 (11.08) |
| Normal | 481(41.97) | |
| Mild | 178(15.53) | |
| Moderate | 143(12.48) | |
| Severe | 344(30.02) | |
| Resilience score | | 82.10 (15.77) |
| Family satisfaction score | | 33.26 (9.27) |

N: number, M(SD): Mean (standard deviation)

after resilience ($\beta$ = -0.26, $p$<0.001) was added in Model 3. Overall the model explains 23% of the variance in depression.

## The mediating role of resilience on anxiety/ depression

Table 4 shows the path coefficients a (family satisfaction to resilience) and b (direct effects of resilience on anxiety/ depression), a*b products (indirect effects of family satisfaction on anxiety/ depression through resilience) and BCa95%CI, coefficients c (total effects of family satisfaction on anxiety/ depression) and c' (direct effects of family satisfaction on anxiety/ depression). For the model, family satisfaction (a = 0.70, $p$<0.001) was positively associated with resilience, and resilience (b = -0.06, $p$<0.001; b = -0.20, $p$<0.001, respectively) was negatively associated with anxiety and depression. The mediating effects of resilience on the

**Table 2. Correlations among major variables.**

| Variables | 1 | 2 | 3 | 4 |
|---|---|---|---|---|
| 1. Family satisfaction | - | | | |
| 2. Resilience | 0.42*** | - | | |
| 3. Anxiety | -0.29*** | -0.28*** | - | |
| 4. Depression | -0.39*** | -0.41*** | 0.72*** | - |

***$p$ < 0.001 (2-tailed)

**Table 3. Hierarchical linear regression analyses for anxiety and depression.**

| | Anxiety | | | Depression | | |
|---|---|---|---|---|---|---|
| Variable | Model 1 ($\beta$) | Model 2 ($\beta$) | Model 3 ($\beta$) | Model 1 ($\beta$) | Model 2 ($\beta$) | Model 3 ($\beta$) |
| Age | 0.05 | 0.03 | 0.03 | 0.07 | 0.05 | 0.05 |
| Sex | 0.07* | 0.07* | 0.05 | 0.12*** | 0.12*** | 0.09*** |
| Grade | 0.01 | 0.00 | -0.01 | -0.03 | -0.04 | -0.05 |
| Housing type | | | | | | |
| Home Ownership Scheme vs Public housing | -0.03 | -0.02 | 0.00 | -0.09** | -0.07* | -0.04 |
| Private housing vs Public housing | -0.08* | -0.05 | -0.03 | -0.11** | -0.07* | -0.04 |
| Others vs Public housing | -0.06 | -0.04 | -0.03 | -0.06* | -0.04 | -0.03 |
| Family satisfaction | | -0.28*** | -0.20*** | | -0.37*** | -0.25*** |
| Resilience | | | -0.18*** | | | -0.26*** |
| F | 2.97 | 15.92 | 18.47 | 5.96 | 32.05 | 42.62 |
| $R^2$ | 0.02 | 0.09 | 0.12 | 0.03 | 0.17 | 0.23 |
| Change in $R^2$ | 0.02 | 0.07 | 0.03 | 0.03 | 0.14 | 0.07 |

*$p < 0.05$, **$p < 0.01$, ***$p < 0.001$

relationship between family satisfaction and anxiety and depression were significant (a*b = -0.04, BCa95%CI: -0.06, -0.03; a*b = -0.14, BCa95%CI: -0.18, -0.11, respectively). The proportions of mediating roles (indirect effect/ total effect) of resilience on anxiety and depression were 26.3% and 31.1%, respectively.

## Discussion

The present study explored the associations between family satisfaction, resilience, and anxiety/ depression among secondary school students in Hong Kong. First of all, it is important to note that 45.8% of our sample scored higher that the cut-off for mild anxiety, which was higher than 22.7% reported among adolescents in the USA [58]. Also, 58.0% scored higher than cut-off for mild depression, which was similar to that reported in a large representative study among secondary school students in Hong Kong (about 60%) [6], but higher than the counterparts in other countries such as mainland China (34.4%) [59] and Australia (38.3%) [60]. Findings were consistent with a previous cross-cultural study that reported a higher level of psychological distress among individuals in Hong Kong than other countries [61]. As Chinese

**Table 4. Mediating role of resilience on the association between family satisfaction and anxiety/ depression.**

| | Path coefficients | | | | a*b (*BCa 95%CI*) |
|---|---|---|---|---|---|
| | C | A | b | c′ | |
| Family satisfaction to anxiety | -0.16*** | 0.70*** | -0.06*** | -0.12*** | -0.04*** (-0.06, -0.03) |
| Family satisfaction to depression | -0.45*** | 0.70*** | -0.20*** | -0.30*** | -0.14*** (-0.18, -0.11) |

*: $p < 0.05$, **: $p < 0.01$, ***: $p < 0.001$

c: total effects of family satisfaction on anxiety/ depression

a: effect of family satisfaction on resilience

b: direct effects of resilience on anxiety/ depression

c′: direct effects of family satisfaction on anxiety/ depression

a*b: indirect effects of family satisfaction on anxiety/ depression through resilience

BCa95%CI: bias-corrected and accelerated 95% confidence interval.

culture emphasizes self-control, obedience and respect for authority, Chinese adolescents may be more vulnerable to developing internalizing pathology [62, 63]. A strong emphasis on academic performance, high levels of parental expectation and control, heavy school workload and intensive extra-curricular activities might also lead to high levels of stress among Hong Kong adolescents [64]. Furthermore, females were more vulnerable to anxiety and depression in our sample, which is consistent with the findings reported in the general population [65–67]. Overall, findings of the present study highlight an urgent need for interventions to reduce the prevalent mental health problems of adolescents in Hong Kong.

Greater family function and relationships are associated lower scores of anxiety and depression among adolescents in the literature. Our results showed that family satisfaction was a protective factor to adolescents' anxiety and depression, which is in line with prior findings [23, 26]. In the present study, the family satisfaction score of our participants was comparable to the study in South America assessed by the same scale (M = 34.2, SD = 6.6) [26]. A review conducted by Bögels & Brechman-Toussaint revealed that poor family functioning such as marital conflict, and co-parenting and parental rearing strategies, predicted child anxiety [68]. Moreover, other studies suggested that integrating family into intervention and treatment of depression among adolescents received greater improvement in functioning, hopelessness and self-reported depression than traditional treatment [69, 70]. Given the critical importance of families in adolescents' mental health, empowering families to enact adaptive changes and involvement in intervention should be considered.

Apart from family satisfaction, our findings also reported that resilience was another protective factor of adolescents' anxiety and depression, which is consistent to the extant literature [31, 33, 71, 72]. Resilience can be viewed as a defense mechanism, which enables people to thrive in the face of adversities [73]. Therefore, resilience may be an essential asset that should be promoted for prevention and treatment of mental health problems.

We found that resilience was associated with family satisfaction, which has not been previously tested in the Chinese context. Such findings corroborate with existing studies that family communication, cohesion and close relationships were positively associated with resilient outcomes in youth among the western population [38, 39, 74]. Furthermore, our results highlighted a significant indirect pathway by which resilience mediated the relationship between family satisfaction and anxiety/ depression. In other words, family satisfaction was associated with higher levels of resilience, which in turn, linked with lower levels of anxiety and depression among adolescents. Based on the bootstrap method, the proportion of mediating roles of resilience on anxiety and depression was 26.3% and 31.1%, respectively. Considering the complexity of mediators between family satisfaction and mental health problems, it is not surprising to find resilience as a partial but not full mediation. To our knowledge, this is the first study to test the potential mediating role of resilience in the effect of family satisfaction on anxiety and depression. Findings offer theoretical support to the dynamic systems perspective of resilience and the Resiliency Theory which posit that social factors, such as family support and family satisfaction, can be useful promotive factors that promote resilience and reduce anxiety/ depression. Adolescents who are satisfied with their family environment might be may be more capable of developing positive coping strategies that can act against the potential harm of stressors and help them manage adversity more effectively. These can further protect them from the detrimental effects of mental illness. The results are also consistent with a study among individuals with multiple sclerosis, which showed that resilience mediated the relationships between social support from family members and subsequent mental health outcomes such as depressive and anxious symptomatology [75]. The importance of engaging family in cultivating resilience and mental health of adolescents is underscored.

## Implications

Findings of the present study would provide important implications for practice. Identifying the mediator of the association between family satisfaction and resilience would be particularly useful as it provides insights on the complex association between the variables, which allows health care professionals to understand the focal point and tailor appropriate strategies for effective mental health promotion. In view of the significant mediating effect of resilience, resilience programs may be a useful intervention in easing adolescents' anxiety and depression symptoms. A randomized control trial conducted in the USA indicated that the resiliency intervention group had significantly higher scores in effective coping strategies, lower scores on depressive symptoms, and less perceived stress than the control group [76]. Similar resiliency interventions also resulted in lower levels of mental health symptoms among populations in Australia and Thailand [77, 78]. These programs promote development of personal psychological attributes such as optimism, self-efficacy and adaptability, and equip adolescents with positive coping methods as a means to promote their resilience and mental health. They also. The mediating role of resilience also suggest that monitoring adolescents' resilience may be a good indicator for health care professors to identify emerging adolescents who are at risk for mental ill health. Resilience should be considered as a viable means to track the changes in mental health of adolescents across time.

Additionally, identification of specific factors that promote resilience may also be effective in promoting mental health among adolescents. The significant association between family satisfaction and anxiety/ depression highlights the importance of involving family members in mental health promotion among adolescents. Interventions that promote family management skills, positive family interaction, and effective family communications would be effective in reducing stress and family conflict and consequently, anxiety and depression of adolescents [79, 80]. Supporting families to create a safe and supportive environment where they can openly discuss their feelings and worries, and providing information and resources about mental health to parents can also help them better understand and support their adolescents in cultivating positive mental health.

## Limitations

There are some limitations in the present study that should be mentioned. First, although we delineated the potential mechanism which resilience and family satisfaction were associated with anxiety and depression, the cross-sectional design precluded the opportunity to draw causal conclusions. Prospective research is warranted to confirm our findings. Second, anxiety and depression were assessed through self-reported questionnaires, rather than clinical diagnosis. However, both the GAD and CES-D are validated and commonly used to screen probable anxiety and depression among adolescents [81, 82]. Third, data was collected from grade 8 and 9 students from 4 secondary schools, findings might not be generalizable to other secondary school students in Hong Kong. Finally, it is acknowledged that other aspects of satisfaction, such as social and life satisfaction, would also have an important influence on adolescent resilience and mental health [83, 84]. The present study only measured the role of family satisfaction on mental health among adolescents. It is a limitation that other aspects of satisfaction have not been captured in the present study.

## Conclusions

In summary, we identified a very high prevalence of probable anxiety and depression among secondary school students in Hong Kong. Family satisfaction has a direct negative association with anxiety and depression and may also strengthen resilience, which in turn, reduce anxiety

and depression symptoms. Both family satisfaction and resilience have instrumental influence on adolescent mental health, therefore, promoting positive family relationships and resilience of adolescents may be an effective strategy in preventing and alleviating anxiety and depression symptoms among adolescents.

## Author Contributions

**Conceptualization:** Joseph T. F. Lau, Phoenix K. H. Mo.

**Data curation:** Joseph T. F. Lau, Phoenix K. H. Mo.

**Formal analysis:** Beizhu Ye.

**Project administration:** Phoenix K. H. Mo.

**Supervision:** Phoenix K. H. Mo.

**Writing – original draft:** Beizhu Ye.

**Writing – review & editing:** Ho Hin Lee, Jason C. H. Yeung, Phoenix K. H. Mo.

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
