## [Decision Letter · Decision Letter 0]

12 Jan 2023

PONE-D-22-33883The mediating role of resilience on the association between family satisfaction and lower levels of depression and anxiety among Chinese adolescentsPLOS ONE

Dear Dr. Phoenix K.H.Mo,

Thank you for submitting your manuscript to PLOS ONE. After careful consideration, we feel that it has merit but does not fully meet PLOS ONE’s publication criteria as it currently stands. Therefore, we invite you to submit a revised version of the manuscript that addresses the points raised during the review process.

We look forward to receiving your revised manuscript.

Kind regards,

Carmen Concerto

Academic Editor

PLOS ONE

Reviewers' comments:

Reviewer's Responses to Questions

**Comments to the Author**

1. Is the manuscript technically sound, and do the data support the conclusions?

Reviewer #1: Partly

Reviewer #2: Partly

2. Has the statistical analysis been performed appropriately and rigorously? 

Reviewer #1: Yes

Reviewer #2: Yes

3. Have the authors made all data underlying the findings in their manuscript fully available?

Reviewer #1: Yes

Reviewer #2: Yes

4. Is the manuscript presented in an intelligible fashion and written in standard English?

Reviewer #1: No

Reviewer #2: Yes

5. Review Comments to the Author

Reviewer #1: ID: PONE-D-22-33883

Title: The mediating role of resilience on the association between family satisfaction and lower levels of depression and anxiety among Chinese adolescents.

Thank you for providing a chance to review this manuscript.

Comment: Major revision.

Detailed information:

Abstract

I would suggest adding some details to the “Methods” section. The “Results” and “Discussion” could be described in more concise language and some non-major results could be removed.

Introduction

Paragraph 1, Page 9: 1) This paragraph is like different sentences put together, the logic is very incoherent; 2) This paragraph is too long, recommend one paragraph for one topic; 3) The background and the discussion are detached. I don’t see Covid-19 in the discussion, but I see a lot in the background, which is unreasonable.

Paragraph 2, Page 10: “On the other hand, their levels of capability in coping with these challenges might not be adequate, which makes them emotionally vulnerable to different mental problems.” ——— It is necessary to add references here.

Paragraph 3, Page 13: What are the assumptions of this study? What is the innovation point and research significance?

The background section is too long, and I think some of the content can be moved to the discussion section, and some of the less relevant content can be deleted. In addition, the logic of the background needs to be strengthened and not give a very strong sense of piecing together.

Methods

Paragraph 1, Page 15: “Cronbach’s alpha = 0.941” ——— Please add references and check if the same problem exists in other parts.

Paragraph 2, Page 16: Which options each variable contains need to be listed.

Paragraph 4, Page 16: The description of the mediating effect is not clear.

Is there quality control in this study? What is the percentage of missing data? What method is used to replace missing values?

Results

Paragraph 4, Page 17: Please don’t just describe the exact contents of the table.

Paragraph 3, Page 18: “The mediating role of resilience in the relationship between family satisfaction and anxiety/ depression” ——— This subtitle is too long

Discussion

What are the advantages of this study？

References

Please double-check the format of the references, I find that many of the formats are inconsistent.

Table

Table 1: Place “(Mean, SD)” in the notes below the table and replace it with a symbol in the table.

Table 3: Any abbreviations appearing in the table must be indicated in full below the form.

Thank you and my best,

Your reviewer

Reviewer #2: This cross-sectional study was conducted among Chinese adolescents to explore the mediating role of resilience on the association between family satisfaction and mental health. This is the first study to test the potential mediating role of resilience in the effect of family satisfaction on anxiety and depression with large sample size. However, there were still many problems in this study that cause me concern.

(1)The introduction should explain why only the aspect of family satisfaction was focused on? What is the importance? It is important to consider that other aspects of satisfaction (eg, social, life) also have an certain impact on resilience and mental health.

(2)There was insufficient longitudinal evidence presented in the introduction section that family satisfaction was a cause of psychological problems. The theories of hypotheses on the mediation role of resilience need to be enriched.

(3)This study included unrepresentative samples of adolescents which may cause huge bias for the results, which is a significant weakness of this study.

(4)How many questionnaires were sent to students in this survey? What was the response rate to the questionnaire? Why only 8th and 9th graders were surveyed?

(5)When was the study investigated? How long is the survey period? Have you considered the impact of the COVID-19 pandemic on this survey?

(6)The discussion section needs to focus on why resilience can mediate the association between family satisfaction on anxiety and depression. What are the underlying mechanisms?

(7)The descriptive of family satisfaction scores should be added in the Table 1.

(8)The implications of this study on public health should be highlighted.

(9)I am confused that why adolescents in Hong Kong have such high levels of anxiety and depression. Can authors give some explanations?

(10)What is the probable anxiety and depression? No such definition was presented in the manuscript.

6. PLOS authors have the option to publish the peer review history of their article (what does this mean?). If published, this will include your full peer review and any attached files.

Reviewer #1: No

Reviewer #2: No

---

## [Author Response · Author response to Decision Letter 0]

13 Feb 2023

Responses Reviewer 1's comments:

Abstract

I would suggest adding some details to the “Methods” section. The “Results” and “Discussion” could be described in more concise language and some non-major results could be removed.

Response: We have followed the reviewer’s comment to include more details to the Methods section and also revise the results and discussion section of the abstract. 

Introduction

Paragraph 1, Page 9: 1) This paragraph is like different sentences put together, the logic is very incoherent; 2) This paragraph is too long, recommend one paragraph for one topic; 3) The background and the discussion are detached. I don’t see Covid-19 in the discussion, but I see a lot in the background, which is unreasonable. 

Response: We have followed the reviewer’s comment to rewrite the introduction. Specifically, those descriptions about COVID-19 were removed and the structure of the whole introduction has been revised to first discuss the importance of mental health and the prevalence of mental health problems among adolescents in Hong Kong, followed by role of family satisfaction and resilience on mental health problems, and the potential mediating role of resilience. We believe that we have produced a revised introduction with better flow, consistency and readability. 

Paragraph 2, Page 10: “On the other hand, their levels of capability in coping with these challenges might not be adequate, which makes them emotionally vulnerable to different mental problems.” ——— It is necessary to add references here. 

Response: We have followed the reviewer’s comment to include a reference that supports the statement. 

Paragraph 3, Page 13: What are the assumptions of this study? What is the innovation point and research significance? 

Response: We have followed the reviewer’s comment to elaborate further on the assumption and the significance of the study in the introduction section:

“to the best of our knowledge, the possible mediating role of resilience in the association between family satisfaction on anxiety and depression has not yet been investigated…. Considering the significant mental health problems documented among Chinese adolescents, there is an urgent need to understand the factors associated with their mental health so that appropriate interventions could be designed. With respect to the indispensable role of family factors and resilience on adolescent mental health, it is expected that findings of the present study would provide important directions and implications on the design of mental health interventions for Chinese adolescents. The present study aimed to explore the association between family satisfaction and resilience on mental health problems (i.e. anxiety and depression) among adolescents in Hong Kong. It also examined whether resilience mediated the relationship between family satisfaction and anxiety/depression. We hypothesized that family satisfaction would be associated with higher levels of resilience, which in turn, would be associated with lower levels of anxiety/depression among adolescents.”

The background section is too long, and I think some of the content can be moved to the discussion section, and some of the less relevant content can be deleted. In addition, the logic of the background needs to be strengthened and not give a very strong sense of piecing together. 

Response: As mentioned before, we have followed the reviewer’s comment to rewrite the introduction. Specifically, those descriptions about COVID-19 were removed and the structure of the whole introduction has been revised. We believe that we have produced a revised introduction with better flow, consistency and readability.

Methods

Paragraph 1, Page 15: “Cronbach’s alpha = 0.941” ——— Please add references and check if the same problem exists in other parts.

Response: We have clarified that all the Cronbach’s alphas were values obtained from the present study. 

Paragraph 2, Page 16: Which options each variable contains need to be listed.

Response: We have followed the reviewer’s comment to list the option under each socio-demographic variable. 

Paragraph 4, Page 16: The description of the mediating effect is not clear.

Response: We have followed the reviewer’s comment to elaborate more on the analysis of the mediating effect in the analysis section: 

“In addition, asymptotic and resampling strategies developed by Preacher and Hayes (Preacher & Hayes, 2008) were employed to test the significance of the mediating role of resilience on the associations between family satisfaction and anxiety/ depression. In the regression equations, family satisfaction was modeled as predictors, anxiety/ depression as the dependent variable, resilience as the mediator, and socio-demographic variables as covariates. The path coefficients a (family satisfaction to resilience), b (direct effects of resilience on anxiety/ depression), a*b products (indirect effects of family satisfaction on anxiety/ depression through resilience), c (total effects of family satisfaction on anxiety/ depression) and c’ (direct effects of family satisfaction on anxiety/ depression) were presented. We applied bootstrapping methods to obtain a 95% confidence interval (CI) for the mediated effects. The bootstrap estimate was based on 5,000 bootstrap samples, and the bias-corrected and accelerated 95% confidence interval (BCa95%CI) of indirect effect (a*b product) excluding 0 indicated a significant mediating role.”

Is there quality control in this study? What is the percentage of missing data? What method is used to replace missing values?

Response: We have followed the reviewer’s comment to elaborate on the quality control, percentage of missing data, and method used to replace missing values: 

“As a measure of quality control, 39 cases with at least one scale having more than 20% of its items missing were excluded from data analysis. The final sample consisted of 1,146 participants.”

“The missing values from all of the items ranged from 0.1% to 0.4%, which were imputed by using the modal value of the other items of the same scale.”

Results

Paragraph 4, Page 17: Please don’t just describe the exact contents of the table.

Response: We have followed the reviewer’s comment to offer some elaborations of the results. 

Paragraph 3, Page 18: “The mediating role of resilience in the relationship between family satisfaction and anxiety/ depression” ——— This subtitle is too long

Response: We have followed the reviewer’s comment to revise the subtitle.

Discussion

What are the advantages of this study？

Response: We have followed the reviewer’s comment to elaborate on the advantages of the study of the present study. Our study is the first to examine the mediating role of resilience on the association between family satisfaction and anxiety/ depression among adolescents in Chinese, such findings have important theoretical and practical implications. 

In terms of theoretical implications, findings offer theoretical support to the dynamic systems perspective of resilience and the Resiliency Theory which posit that social factors, can be useful promotive factors that promote resilience and reduce anxiety/ depression. It also underscored the importance of engaging family in cultivating resilience and mental health of adolescents. 

In terms of practical implications, delineating the mediating variables on the association between family satisfaction and anxiety/ depression provides insights on the complex association between the variables, which allows health care professionals to understand the focal point and tailor appropriate strategies for effective mental health promotion. In view of the significant mediating effect of resilience, resilience programs may be a useful intervention in easing adolescents’ anxiety and depression symptoms. We have further elaborated these and provided some examples of successful interventions in promoting resilience and family satisfaction in the implications section. 

References

Please double-check the format of the references, I find that many of the formats are inconsistent.

Response: We have followed the reviewer’s comment to edit the format of the references. 

Table

Table 1: Place “(Mean, SD)” in the notes below the table and replace it with a symbol in the table.

Response: We have followed the reviewer’s comment to place “(Mean, SD)” in the notes below the table and replace it with a symbol in the table.

Table 3: Any abbreviations appearing in the table must be indicated in full below the form.

Response: We have followed the reviewer’s comment to indicate the full name of all abbreviations appearing in the table. 

Responses to Reviewer 2’s comments

This cross-sectional study was conducted among Chinese adolescents to explore the mediating role of resilience on the association between family satisfaction and mental health. This is the first study to test the potential mediating role of resilience in the effect of family satisfaction on anxiety and depression with large sample size. However, there were still many problems in this study that cause me concern.

(1)The introduction should explain why only the aspect of family satisfaction was focused on? What is the importance? It is important to consider that other aspects of satisfaction (eg, social, life) also have an certain impact on resilience and mental health. 

Response: We have followed the reviewer’s comment to elaborate further the rationale of focusing on family satisfaction in the introduction section. We focused on family satisfaction as it is important to examine social factors to adolescents’ mental health and family satisfaction is one of the most influential one among the social factors. 

“Studies have shown that adolescents’ anxiety and depression are subject to the influence of various individual and social factors. (Cheng et al., 2014; Lamb et al., 2010; Mathews et al., 2016; Niarchou et al., 2015; Rueger et al., 2016). Among them, the influence of family on adolescent mental health is instrumental (Carr, 2006; Kienhorst, 2000; Rohner & Britner, 2002). Specifically, family satisfaction, which measures the extent to which one is satisfied with one’s family’s overall functioning and relationships, is one of the most influential aspects of life satisfaction. Studies have shown that higher family satisfaction leads to lower levels of worry, anxiety, and depression in adolescents (Proctor & Linley, 2014)…”

We agree with the reviewer that other aspects of satisfaction would also have an important influence on resilience and mental health. It is a limitation that these have not been measured in the present study. We have elaborated this in the limitation section. 

(2)There was insufficient longitudinal evidence presented in the introduction section that family satisfaction was a cause of psychological problems. The theories of hypotheses on the mediation role of resilience need to be enriched. 

Response: We have followed the reviewer’s comment to include more evidence of the role of family satisfaction on anxiety and depression from longitudinal studies:

“Both cross-sectional and longitudinal studies have proven that family satisfaction was a significant protective factor of anxiety and depression among adolescents (Magson et al., 2021; Vulic-Prtoric & Macuka, 2006; Way & Robinson, 2003). One study in South America found that family satisfaction and communication explained 18% of the variance in child anxiety (Nicolais et al., 2016).”

We have also included the Resiliency Theory, that could provide support to the mediation role of resilience on mental health in the introduction section:

“According to the Resiliency Theory, two types of promotive factors operate to ameliorate risk factors and mental ill health: positive factors within individuals and resources available outside the individuals (Fergus & Zimmerman, 2005; Zimmerman et al., 2013). In this regard, family satisfaction could serve as an important positive resource that promotes resilience, which further buffer the negative effects of stress of mental illness.”

(3)This study included unrepresentative samples of adolescents which may cause huge bias for the results, which is a significant weakness of this study.

Response: We agree with the reviewer that sampling bias might exist in the present study, we have included this in the limitation section: 

“data was collected from grade 8 and 9 students from 4 secondary schools, thus findings might not be generalizable to other secondary school students in Hong Kong.”

(4)How many questionnaires were sent to students in this survey? What was the response rate to the questionnaire? Why only 8th and 9th graders were surveyed?

Response: We have clarified that all grade 8 and 9 students from the selected schools were invited, and all of the returned the questionnaire. We also provided explanation on why only 8th and 9th graders were invited. 

“An opt-out parental informed consent was used, in which parents were asked to return a signed form if they refused to let their students take part in the survey. No parents returned such forms……The questionnaire was sent to a total of 1,185 students, and all of them returned the questionnaire.”

“Discussions were made with the school management team, it was agreed that secondary two and three students (i.e. grade eight and nine) would be invited as secondary one students have just transited to secondary school, and secondary four to six students would need to prepare for public examinations.” 

(5)When was the study investigated? How long is the survey period? Have you considered the impact of the COVID-19 pandemic on this survey?

Response: We clarified that the study was conducted between January to March 2017 and the survey took around 20 minutes to complete. 

(6)The discussion section needs to focus on why resilience can mediate the association between family satisfaction on anxiety and depression. What are the underlying mechanisms? 

Response: We have followed the reviewer’s comment to elaborate more on the underlying mechanisms which resilience might mediate the association between family satisfaction on anxiety and depression in the discussion section: 

“Findings offer theoretical support to the dynamic systems perspective of resilience and the Resiliency Theory which posit that social factors, such as family support and family satisfaction, can be useful promotive factors that promote resilience and reduce anxiety/ depression. Adolescents who are satisfied with their family environment might be may be more capable of developing positive coping strategies that can act against the potential harm of stressors and help them manage adversity more effectively. These can further protect them from the detrimental effects of mental illness.”

(7)The descriptive of family satisfaction scores should be added in the Table 1.

Response: We have followed the reviewer’s comment to include the mean score of family satisfaction and resilience in Table 1.

(8)The implications of this study on public health should be highlighted. 

Response: We have followed the reviewer’s comment to include the implications of the study in the discussion section: 

“Findings of the present study would provide important implications for practice. Identifying the mediator of the association between family satisfaction and resilience would be particularly useful as it provides insights on the complex association between the variables, which allows health care professionals to understand the focal point and tailor appropriate strategies for effective mental health promotion. In view of the significant mediating effect of resilience, resilience programs may be a useful intervention in easing adolescents’ anxiety and depression symptoms. A randomized control trial conducted in the USA indicated that the resiliency intervention group had significantly higher scores in effective coping strategies, lower scores on depressive symptoms, and less perceived stress than the control group (Steinhardt & Dolbier, 2008). Similar resiliency interventions also resulted in lower levels of mental health symptoms among populations in Australia and Thailand (Grant et al., 2009; Songprakun & McCann, 2012). These programs promote development of personal psychological attributes such as optimism, self-efficacy and adaptability, and equip adolescents with positive coping methods as a means to promote their resilience and mental health. They also. The mediating role of resilience also suggest that monitoring adolescents’ resilience may be a good indicator for health care professors to identify emerging adolescents who are at risk for mental ill health. Resilience should be considered as a viable means to track the changes in mental health of adolescents across time. 

Additionally, identification of specific factors that promote resilience may also be effective in promoting mental health among adolescents. The significant association between family satisfaction and anxiety/ depression highlights the importance of involving family members in mental health promotion among adolescents. Interventions that promote family management skills, positive family interaction, and effective family communications would be effective in reducing stress and family conflict and consequently, anxiety and depression of adolescents (Kuhn & Laird, 2014; Kumpfer & Magalhães, 2018). Supporting families to create a safe and supportive environment where they can openly discuss their feelings and worries, and providing information and resources about mental health to parents can also help them better understand and support their adolescents in cultivating positive mental health.”

(9)I am confused that why adolescents in Hong Kong have such high levels of anxiety and depression. Can authors give some explanations?

Response: We have followed the reviewer’s comment to provide explanations on the high levels of anxiety and depression among adolescents in Hong Kong in the discussion section: 

“As Chinese culture emphasizes self-control, obedience and respect for authority, Chinese adolescents may be more vulnerable to developing internalizing pathology (Constantine et al., 1997; Sun & Rao, 2017). A strong emphasis on academic performance, high levels of parental expectation and control, heavy school workload and intensive extra-curricular activities might also lead to high levels of stress among Hong Kong adolescents (Shek DTL., 2014).”

(10)What is the probable anxiety and depression? No such definition was presented in the manuscript.

Response: We have followed the reviewer’s comment to clarify that probable anxiety and depression refers to those who scored above the cut-off for mild anxiety or mild depression.

---

## [Editor Report · Decision Letter 1]

14 Mar 2023

The mediating role of resilience on the association between family satisfaction and lower levels of depression and anxiety among Chinese adolescents

PONE-D-22-33883R1

Dear Dr. Phoenix k. H. Mo,

We’re pleased to inform you that your manuscript has been judged scientifically suitable for publication and will be formally accepted for publication once it meets all outstanding technical requirements.

Kind regards,

Carmen Concerto

Academic Editor

PLOS ONE
---

## [Editor Report · Acceptance letter]

17 May 2023

PONE-D-22-33883R1 

The mediating role of resilience on the association between family satisfaction and lower levels of depression and anxiety among Chinese adolescents 

Dear Dr. Mo:

I'm pleased to inform you that your manuscript has been deemed suitable for publication in PLOS ONE. Congratulations! Your manuscript is now with our production department. 

Kind regards, 

on behalf of

Dr. Carmen Concerto 

Academic Editor

PLOS ONE